# Genetic Diversity and Identification of *Vaccinium* Species Through Microsatellite Analysis

**DOI:** 10.3390/plants13243488

**Published:** 2024-12-13

**Authors:** Márcia Carvalho, Manuela Matos, António Crespí, Violeta R. Lopes, Valdemar Carnide

**Affiliations:** 1Centre for the Research and Technology of Agro-Environmental and Biological Sciences (CITAB), University of Trás-os-Montes e Alto Douro (UTAD), 5000 Vila Real, Portugal; mmatos@utad.pt (M.M.); acrespi@utad.pt (A.C.); vcarnide@utad.pt (V.C.); 2Institute for Innovation, Capacity Building and Sustainability of Agri-Food Production (Inov4Agro), University of Trás-os-Montes e Alto Douro (UTAD), 5000 Vila Real, Portugal; 3Portuguese Genebank of Vegetal Germplasm (BPGV), National Institute of Agrarian and Veterinary Research, 4700 Braga, Portugal; violeta.lopes@iniav.pt

**Keywords:** *Vaccinium corymbosum*, *Vaccinium myrtillus*, *Vaccinium ashei*, simple sequence repeats, genetic diversity

## Abstract

The *Vaccinium* genus contains about 500 species distributed worldwide but only a limited number of species have been studied for genetic diversity using molecular markers. In this study, a genetic analysis was conducted on three Vaccinium species (four cultivars of *V. corymbosum*, four wild populations of *V. myrtillus*, and two cultivars of *V. ashei*), for a total of 95 genotypes, using eight microsatellite (SSR) loci. A total of 57 alleles were detected. The number of alleles per locus ranged from 2 to 14, with an average of 7.25. Six unique alleles in *V. corymbosum*, four in *V. ashei*, and three in *V. myrtillus* were identified as being potential species markers. The dendrogram and principal coordinate analysis revealed a clear division of the three species into distinct groups, with each group further divided into sub-clusters based on the type of cultivars and population origin. The set of SSR primers used in this study demonstrated cross-species transferability, allowing their utilization in *V. ashei* and *V. myrtillus*, and can be used as a reliable tool for cultivar/population and species identification in blueberries.

## 1. Introduction

*Vaccinium* L. is a member of the Ericaceae family, comprising about 500 acid-loving species distributed worldwide, except in Antarctica and Australia [1]. Blueberry (*Vaccinium* section Cyanococcus) is an important small fruit crop, well-adapted to acidic and sandy soils, which has been harvested in Europe for centuries [2,3]. Blueberries were domesticated more recently than other *Vaccinium* species. Three blueberry species are considered commercially significant and are mainly responsible for global blueberry production: *Vaccinium corymbosum* L. (highbush blueberry), *V. ashei* R. (rabbiteye blueberry, syn. *V. virgatum*), and *V. angustilolium* Ait. (lowbush blueberry) [4].

The highbush blueberry (2n = 4x = 48) is one of the most economically significant *Vaccinium* species globally. Originating in North America, it has become a significant agricultural commodity, with extensive production in the United States, Canada, and Europe. Highbush blueberries are consumed in their fresh, frozen, or processed forms, including as juices, jams, and baked goods [5,6].

*V. ashei* (2n = 6x = 60), commonly known as rabbiteye blueberry, is a blueberry species native to the Southeastern United States [6]. Its commercial and economic importance is related to several factors, including its adaptability to southern climates, high market demand for both fresh and processed products, export potential, and contribution to the health food market. The high yield, long-term profitability for farmers, and positive environmental contributions make it a key crop in the agricultural industry, particularly in regions where other blueberry varieties may not thrive [6,7].

*V. myrtillus* L. (commonly referred to as bilberry or wild blueberry; 2n = 2x = 24) is a perennial wild shrub found in acidic soils, ranging from mountainous mineral heaths to forest organic soils and old peat bogs across Northern to Central Europe [8]. This crop is suffering a biogeographic regression due to climate change and the thermal recovery occurring on the planet, which may, paradoxically, make it more resilient [9]. Its fruit is small and dark blue but it is not as widely cultivated as other *Vaccinium* species. Wild blueberries have significant commercial applications, particularly in Europe, where it is used in medicinal products and as a food ingredient. This crop is valued for its high levels of anthocyanins, flavonoids, and other polyphenolic compounds, which contribute to its reputation as a popular ingredient in dietary supplements, functional foods, and traditional medicine. Wild blueberries are challenging to cultivate as they grow slowly, are prone to disease, and produce fewer berries compared to cultivated blueberries [10,11]. In recent years, the demand for and the consumption of blueberries has grown mainly due to their health benefits, including a reduced risk of heart disease, chronic diseases, cancer, and the treatment of urinary conditions and diseases. These health-promoting effects are largely attributed to their high levels of phenolic compounds, particularly flavonoids [12,13].

Traditional methods for cultivar identification and characterization in blueberries rely on morphological traits, which are heavily influenced by environmental conditions and production practices. Additionally, collecting these traits can be time-consuming, especially when surveying large populations across different locations [14]. DNA-based molecular markers, however, offer a powerful tool for differentiating among plant genetics resources and for genetic identification and characterization of cultivars since they show high levels of polymorphisms and are not affected by environmental factors [15]. Specifically, molecular markers could lead to a rapid genetic improvement of blueberries and facilitate accurate genotype identification, providing valuable support to breeding programs.

A number of molecular markers have been developed for use in different crops, with applications including species and cultivar identification, genetic diversity analysis, assessment of genetic relationships, genotype fingerprinting, and quality control of rootstock-seed lots [14,16]. Due to their high polymorphism, reproducibility, and codominant nature, microsatellites (simple sequence repeats, SSRs) have become the markers of choice for compiling, standardizing, and exchanging information related to the genetic resources of different species [17].

Several genetic studies have been carried out in the *Vaccinium* genus with randomly amplified polymorphic DNA (RAPD) and inter-simple sequence repeat (ISSR) primers for mapping, genetic diversity analysis, and the DNA fingerprinting of cultivars [4,18,19,20,21]. Furthermore, expressed sequence tags (ESTs) have been utilized for the identification of cultivars in certain *Vaccinium* species [22,23,24,25], and the analysis of genetic diversity and cultivar identification through the use of microsatellite markers (SSRs) has also been reported [1,6,14,26,27,28,29]. In 2005, Boches et al. [26] developed and isolated 30 SSR loci obtained from expressed sequence tags (ESTs) and genomic libraries in *V. corymbosum*; these markers have been successfully used to assess genetic diversity, to characterize genotypes, and to identify cultivar and genotypic variation among tetraploid and diploid *V. corymbosum* accessions [1,6,14,30,31].

Despite their advantages, studies developed to determine genetic relationships between highbush blueberry cultivars and different *Vaccinium* species are rare, contrary to other species where molecular characterization is the basis for classification.

The objectives of this study were to (i) use SSR primers designed for *V. corymbosum* in *V. ashei* and *V. myrtillus* to confirm the cross-species transferability; (ii) assess the levels and patterns of genetic variability among a representative sample of *Vaccinium* cultivars/populations; and (iii) use SSR markers for the characterization and evaluation of genetic diversity and the relationships between three *Vaccinium* species.

## 2. Results

Eight SSR loci analyzed produced a total of 57 different alleles across the 95 genotypes (Table 1). The number of alleles per locus ranged from 2 (CA190R locus) to 14 (CA483F locus) with an average of 7.125 (Table 1). A total of 14 unique alleles were detected, with the CA483F and CA94F loci producing the highest number of exclusive alleles, while, in contrast, the CA190R locus did not present any. In this data set, seven rare alleles were identified, namely the CA94F locus which produced the highest number of alleles (three). The polymorphic information content (PIC) varied between 0.28 (VCC_B3 locus) and 0.37 (CA483F locus), with an average of 0.32 (Table 1).

To explore the data and identify possible species/cultivar/population markers, an analysis was performed exploring the alleles in each of them. The number of different alleles/loci detected ranged from 10 in the Alvão Natural Park population of *V. myrtillus* to 23 in the “Bluecrop” and ”Ozarkblue” cultivars both of *V. corymbosum*, with a total of 149 alleles for the 95 genotypes (Table 2) and an average of 16.9 alleles per locus. The *V. myrtillus* Marão Mountain (MA) population had the highest effective number of alleles (1.079) while the *V. ashei* ”Powderblue” cultivar and *V. corymbosum* ”Duke” and “Goldtraube” cultivars showed the lowest value (1.000), with an average of 1.030 for the three species (Table 2).

Comparing the three *Vaccinium* species, six unique alleles in *V. corymbosum* (CA23F, CA112F, CA94F, CA483F, and VCC_B3 loci), four in *V. ashei* (CA23F, CA112F, CA169F, and CA94F loci) and three in *V. myrtillus* (CA112F and VCC_I8 loci) were identified.

The eight SSR loci allowed the identification of two unique alleles in the ”Goldtraube” cultivar (CA169F and CA483F loci) and one in the ”Bluecrop” cultivar (VCC_3 loci), six exclusive alleles (two alleles each in the CA23F, CA483F, and CA94F loci) in ”Powderblue” and three alleles (CA112F, CA94F, and VCC_I8 loci) in the “Ochlockonee” cultivar. The analysis of *V. myrtillus* populations no unique allele was detected as a potential marker for the species population.

The analysis of molecular variance (AMOVA), performed with the ten cultivars/populations of the three *Vaccinium* species, led to a PhiPT value of genetic variation between species of 0.949 (*p* < 0.0001), which indicates a great genetic differentiation (Table 3). Molecular variance was 95% among populations (the included populations/cultivars/species), while within populations there was 5%, indicating that there is more variation among populations/cultivars/species than within populations.

The dendrogram (Figure 1) generated to determine the genetic relationship among the three *Vaccinium* species revealed a similarity coefficient ranging from 0.52 to 1.0, which allows for clear distinction and separation of the three *Vaccinium* species. The first cluster (I) includes the 40 genotypes of *V. myrtillus* populations (sub-cluster I.1) and the 15 genotypes of the two cultivars of *V. ashei* (sub-cluster I.2). The sub-cluster I.1. can be divided into two sub-clusters with a coefficient of similarity of 0.86 (I.1.1 and I.1.2). Sub-cluster I.1.1 grouped all 20 genotypes from the wild population in the Marão and Alvão mountains, areas belonging to the Natura 2000 network, while sub-cluster I.1.2 contains the 20 genotypes from the Peneda-Gerês National Park (PGNP). Sub-cluster I.2 has a coefficient of similarity of 0.78. The ”Ochlockonee” genotypes are grouped in sub-cluster I.2.1, while the ”Powderblue” genotypes form sub-cluster II.2.2. The second cluster (II) includes all the *V. corymbosum* cultivars, with a similarity coefficient ranging from 0.75 to 1.0. The ”Bluecrop”, ”Duke”, and ”Ozarkblue” cultivars are grouped in sub-cluster II.1, while the ”Goldtraube” cultivar is in sub-cluster II.2 with a coefficient of similarity of 0.75.

The results of the principal coordinate analysis (PCoA) (Figure 2) align with the findings of the hierarchical analysis. All *V. myrtillus* populations form one isolated group, while *V. corymbosum* cultivars cluster into a second group, and *V. ashei* cultivars form a third group. The first two principal coordinates of individual genotypes account for 72.17% of the total genetic variance, with coordinate 1 contributing 49.42% and coordinate 2 contributing 22.76%.

The three *Vaccinium* species (a total of 95 genotypes) were further evaluated for population stratification using the STRUCTURE software version 2.3.4. SSR data were analyzed by increasing the number of subpopulations (K) from 1 to 7. The estimation of ΔK, using the Structure Harvester, revealed the highest value for K = 3 (ΔK = 3.014), indicating the presence of three groups primarily composed of the three *Vaccinium* species (Figure 3). Consequently, we selected K = 3 based on the ad hoc ln Pr(X|K) method [32], which recommends picking the smallest value of K that captures the major structure of the data. The *V. myrtillus* population from Marão Mountain was identified as having little mixed ancestry with *V. ashei*.

## 3. Discussion

An accurate identification and characterization of different genotypes and species is fundamental for cultivar development, certification, and conservation but it is also important in the propagation, farming/cultivation, and promotion of commercial cultivars and species.

In the eight SSR loci analyzed, 57 different alleles across the 95 genotypes were detected (Table 1). The average number of alleles per locus was 7.125, the lowest number was 2, and the highest number was 14. The 100% percent polymorphism (in the set of the three species that were not identified as alleles present in all the samples) observed for each SSR locus can be due to the high level of the polymorphism characteristic of this marker [33] and the use of three different species. In *Ficus carica* L. [34] and *Solanum tuberosum* L. [35], for example, a similar result was also verified. Comparing the number of total alleles in this study with the values obtained by Boches et al. [26], a higher number of total alleles in each similar locus was observed, namely in the CA112F locus (11/5), and in the CA483F locus (14/9). The different number of alleles obtained could be explained by the different number of species analyzed. In this study, a total of 149 alleles for the 95 genotypes and an average of 16.9 alleles per locus were observed. Boches et al. [27] in *V. corymbosum* and Hirai et al. [31] in *V. sieboldii* and *V. ciliatum* observed a high number of alleles, with the average number of alleles per locus being 17.7 and 23.4, respectively. PIC values revealed that the CA483_F locus (0.37) is the most informative, and this result is in concordance with Bhatt and Debnath [1]. In contrast, theVCC_B3 locus exhibited the lowest level of informativity (0.28). The unique alleles could be very useful for accurate species/cultivar/population identification, allowing investigators and/or producers to avoid possible errors during production since they are considered as specific markers [20]. This set of SSR markers allowed the identification of unique alleles specific to each Vaccinium species: six unique alleles in *V. corymbosum*, four in *V. ashei*, and three in *V. myrtillus*. Consequently, these alleles could be very useful for accurate species identification and characterization and at the same time, can be incorporated into breeding programs.

Besides the species, we also identified twelve unique alleles in *V. corymbosum* and *V. ashei* cultivars; however, no exclusive allele was detected in *V. myrtillus* populations using the eight SSR loci. The Portuguese populations of *V. myrtillus* are isolated with low genetic diversity at the limit of the range of distribution. They are currently facing biogeographic regression [9] as well as displaying a lack of taxonomic diversity to promote genetic flow [36]. This is a situation not exclusive to this species, as similar trends have been observed in other taxa where populations have exhibited insufficient divergence to allow for the identification of unique alleles [37,38]. These populations (ANP, MA, PGNP-TB, and PGNP-M) occur under lower supratemperate thermotypes [39] but these clearly separated between them (ANP and MA at the eastern, and PGNP-TB and PGNP-M at the western). In this sense, the genetic isolation is explained by the disruptive geomorphology of this region, on the occidental edge of the northern Iberian mountain systems [40]. The clustering obtained by hierarchical analysis and principal coordinate analyses allows for the identification and clear separation of the three *Vaccinium* species. All genotypes of *V. myrtillus* populations are included in sub-cluster I.1, which is divided into two sub-clusters. Sub-cluster I.1.1 contains the genotypes from the Marão (MA) and Alvão (ANP) mountains. A clear separation between the ANP and MA mountain populations was observed. Although the climatic conditions that characterize the localities where the two populations were collected are very similar, this separation could be explained by other characteristics such as substrate, altitude, and orientation. Shale substrate, lower altitude (700 m), and a northern orientation characterize the environmental conditions of the MA population. Conversely, granitic substrate, higher altitude (1100 m), and a western orientation describe the environmental conditions of the ANP population. Sub-cluster I.1.2 contains 20 genotypes from Peneda-Gêres National Park (PGNP). The Melgaço populations (PGNP-M) are from a higher altitude (950 m), lower annual temperature (9 °C), and lower annual temperature range (11 °C), with annual precipitation around 2500 mm. On the other hand, the Terras de Bouro populations (PGNP-TB) are from a region of lower altitude (700 m), higher annual temperature (12 °C), and higher annual temperature range (13 °C), with annual precipitation greater than 3000 mm. Therefore, the separation/differentiation of these populations may be related to the environmental conditions, which appear to be important for the differentiation and creation of genetic diversity in blueberries. These findings are also significant in the context of natural resource conservation, particularly in situ conservation. Despite apparent similarities in their macro-level origins, populations exhibit genetic variability that gives them distinct profiles. In situ conservation strategies must, therefore, consider the implications of these results for appropriate conservation measures when managing natural plant resources and populations of this species in protected areas and natural habitats [41,42,43]. The sub-cluster I.2 includes the genotypes of the two cultivars of *V. ashei* and the ”Ochlockonee” and ”Powderblue” genotypes, which are grouped separately into two sub-clusters (I.2.1 and I.2.2, respectively). Both cultivars are varieties for the late season but have different morphological characteristics; ”Powderblue” has a sky-blue color, mild flavor, is medium-sized and is upright, while ”Ochlockonee” has a medium-blue color, good sweet flavor, a larger size, and is moderately upright. These differences could be responsible for the separation of the samples in the dendrogram. The ”Ochlockonee” cultivar is more recent and resistant to bacterial canker than other rabbiteyes.

The second cluster (II) includes the samples of *V. corymbosum* cultivars, and a low genetic variability (0.75 and 1.0 coefficient of similarity) was observed between the four cultivars of this species. The low genetic variability can be explained by the vegetative propagation and a consequently restricted gene pool that has been used in highbush blueberry breeding programs. Some reports point to the spontaneous *in vivo* mutations that occur related to ploidy or to the method of obtaining hybrids as possibly responsible for the differences between cultivars produced by vegetative propagation [25,28]. This cluster is divided into two sub-clusters: one with the ”Bluecrop”, ”Duke”, and ”Ozarkblue” cultivars (II.1) and the other with ”Goldtraube” (II.2). This distance probably occurs due to the European origin of the ”Goldtraube” cultivar in comparison to the others cultivars, which have an American origin. The same separations were verified in *V. myrtillus* in a study performed by Bassil et al. [44] with the European and American samples grouped into two different clusters. The observed similarities between samples from ”Duke” and ”Ozarkblue” cultivars can be attributed to misclassification of the cultivars during commercialization process.

The *V. ashei* and *V. myrtillus* species are closer than *V. corymbosum*, contrary to what would be expected, probably because *V. ashei* is considered a segmental autoallopolyploid by ancestry [3,10] while *V. corymbosum* is autotetraploid. Another hypothesis is that *V. corymbosum* began to be cultivated in the 1900s and *V. ashei* only started to be cultivated in the 1940s [3]; consequently, as *V. ashei* breeding is a recent event, the separation between them is larger.

The three analyses conducted in this study (dendrogram, PCoA, and STRUCTURE) presented comparable results, with the three *Vaccinium* species exhibiting distinct separation and the cultivars/populations of each species being grouped together.

SSR markers are mainly used for genetic studies within a single species but several studies have demonstrated the efficacy of cross-species transferability of SSRs [45,46,47,48]. For example, Wünsch and Hormaza [45] performed a study where 34 SSR loci, previously developed in peach (*Prunus persica* L.), were used in sweet cherry (*Prunus avium* L.) genotypes. Beyond the efficacy of cross-species transferability of SSR sequences in this study, they also allowed for the differentiation of blueberry genotypes.

As has already been mentioned, the set of eight SSR primer pairs selected for this study were amplified successfully in the three blueberry species and, consequently, proved the transferability of SSRs. Other studies have been developed in different species of the *Vaccinium* genus using the SSR primers developed by Boches et al. [26]; however, at the moment, as far as we know, they have not been tested in *V. ashei*. Boches et al. [27] tested 36 SSR primer pairs in different species of *Vaccinium* genus and concluded that the SSR primers developed in tetraploid *V. corymbosum* were most easily transferable to other members of the section Cyanococcus and least to the sections Oxycoccus, Herpothamnus, Myrtilius, and Batodendron. In Bassil et al. [30]—with SSR primer fingerprinted 16 *V. macrocarpon* (cranberry) cultivars—and in 2010, Bassil et al. [44] identified three *V. reticulatum* clones. These findings suggest that SSR can be used in several species of *Vaccinium* in the section *myrtillus*.

## 4. Materials and Methods

### 4.1. Plant Material

The analysis was conducted on three *Vaccinium* species: two cultivated, (*V. corymbosum* and *V. ashei*), and one wild (*V. myrtillus)*. Each cultivated species comprised four (“Bluecrop”, “Duke”, ”Goldtraube”, and “Ozarkblue”) and two (“Ochlockonee” and “Powderblue”) cultivars, respectively. The *V. myrtillus* species was observed in four distinct populations, two of which were situated within the Natura 2000 network, specifically Alvão Natural Park (ANP) Mountain and Marão Mountain (MA). The remaining two populations were located within Peneda-Gêres National Park, specifically in the Melgaço region (PGNP-M) and the Terras de Bouro region (PGNP-TB). A total of ten genotypes from each cultivar/population and from different individual plants were selected, except for the ”Powderblue” cultivar, with five genotypes. The genotypes were stored apart from each other for more than 200 m, to guarantee a lack of recent kinship. The *V. corymbosum* and *V. ashei* cultivars were from an organic plantation in the central region of Portugal. The geographic origins of the cultivars/populations are presented in Table 4.

### 4.2. DNA Extraction

Young leaves were collected from healthy individuals, frozen in liquid nitrogen at the site, and then stored at −80 °C until DNA extraction.

DNA extraction was performed using the NucleoSpin^®^ Plant II kit (Macherey-Nagel, Düren, Germany) using the Lysis Buffer PL1 and the standard protocol according to the manufacturer’s instructions, with minor modifications in incubation time (75 min) and extra centrifugation, as previously described in [19]. The purified total DNA was quantified and its quality verified by spectrophotometry with NanoDrop 2000 (Thermo Fisher Scientific, Waltham, MA, USA) and by agarose gel electrophoresis, 1% (*w*/*v*). Total DNA samples were stored at −20 °C.

### 4.3. SSR-PCR

Ten previously described SSR loci (CA23F, CA112F, CA169F, CA190R, CA483F, CA787F, CA94F, VCC_B3, VCC_H9, VCC_I8) [26] were tested. Based on the clear banding patterns and size of the amplification products, eight were selected to carry out the analysis (Table 5); the forward primers were labeled with fluorescent tags FAM, NED, PET, or VIC.

PCR amplifications were performed in a 10 µL volume with Qiagen Multiplex PCR Kit (Qiagen, Hilden, Germany) containing 5 μL of Taq master mix, 2 μL DNA template (~30 ng), and 1 μL of SSR mixed primers (a final concentration of 0.15 μM of each primer).

DNA amplifications were carried out in a Biometra UNO II-thermoblock thermocycler (Biometra, Göttingen, Germany) using the following cycle profile: initial denaturation at 94 °C for 3 min, followed by 35 cycles of 94 °C for 30 s, 62 °C for 45 s, 72 °C for 2 min, and a final extension at 72 °C for 30 min. The validation of the PCR amplifications was performed by electrophoresis on agarose gels, 2% (*w*/*v*), running at 90 V for 75 min and stained in an ethidium bromide solution. The gels were visualized using the Molecular Image Gel-Doc^TM^ XR^+^ with Image Lab^TM^ software version 1.0 (BioRad, Hercules, CA, USA). After the validation of all PCR amplifications, PCR products were diluted to equalize the fluorescence intensity, added to 0.2 μL of LIZ 500 size standard, and separated by capillary electrophoresis using an ABI Prism 3730 DNA analyzer (Thermo Fisher Scientific, Waltham, MA, USA). SSR data were analyzed with Peak Scanner software version 1.0 (Thermo Fisher Scientific, Waltham, MA, USA).

### 4.4. Data Analysis

Due to the multiple ploidy levels of the blueberry species, each SSR peak was scored as present (1) or absent (0) for the different cultivars/populations [1]. The number of alleles, in the correct size range, in each cultivar was counted per locus. Alleles were scored by fitting peaks into bins encompassing less than one nucleotide, and peaks significantly smaller than the tallest peak were not scored [14]. The numbers of total, polymorphic, unique, and rare alleles, and the polymorphic information content (PIC) were determined. The alleles were considered unique when a determined allele was detected in only one cultivar/population, while rare alleles were detected in no more than 5% of the samples. The PIC values indicate the marker’s ability to detect polymorphism within the population [1] and were calculated using the program Online Marker Efficiency Calculator (iMEC) [48]. The number of alleles per locus (Na) and the number of effective alleles (Ne) were calculated using GenAlEx software version 6.5 [49].

Principal coordinate analyses (PCoA) were performed based on pairwise genetic similarity distances as estimated by Eigen procedure of GenAlEx 6.5 [50] in order to visualize and analyze the genetic relationships among and within populations/cultivars and species. An analysis of molecular variance (AMOVA) was performed using GenAlEx 6.5 software [50] in order to estimate the genetic variation within and among the species and populations. AMOVA was performed using Phi Statistics (PhiPT) and using 9999 bootstrap iterations. PhiPT statistic is used with binary data sets and is analogous to FST estimates in line with Weir and Cockerham to quantify genetic diversity among populations [50]. For the cluster analysis, applying an un-weighted pair-group method with arithmetic averages (UPGMA) based on a simple matching similarity matrix and SAHN subroutine in NTSYS-pc software version 2.02 was used. Cultivar and population structure and identification of admixed individuals were performed using the model-based software program STRUCTURE 2.3 version 2.3.4. This software uses a Markov chain Monte Carlo (MCMC) algorithm to cluster individuals into populations on the basis of multi-locus genotype data [50]. The number of populations (K) was estimated by performing at least five independent runs of STRUCTURE, using 1,000,000 MCMC repetitions and 50,000 burn-in periods by setting K from 1 to 7. Any prior information about the population of origin was used, and correlated allele frequencies and admixture were assumed. The average of the log-likelihood estimates for each K was calculated. The ad hoc statistic ΔK [51] was used to set the number of populations (K).

## 5. Conclusions

In conclusion, the set of SSR loci used in this study and developed for highbush blueberry (*V. corymbosum*) are transferable to rabbiteye blueberry (*V. ashei*) and bilberry (*V. myrtillus*). This set of primers was sufficient to differentiate and characterize the three *Vaccinium* species. Specific positive markers were detected, which could be very useful for identifying species and cultivars/populations and, consequently, help in the management of germplasm collection—ex situ and in situ—and breeding programs. *V. myrtillus* and *V. ashei* are genetically closest and *V. corymbosum* is more distant. Our results suggest that environmental conditions and localization seem to have some importance in the genetic differentiation of wild *V. myrtillus* populations since they were grouped according to their geographical origin. There was a clear separation between USA-origin cultivars and the ones of European origin, which shows that they have different gene pools.

As has been referred to previously, the identification of blueberry varieties using morphological traits is difficult, and it remains necessary to promote new approaches for reliable and rapid identification. This set of SSR loci provides an opportunity for this screening, allowing producers identification using only plant leaves. For this reason, in the future, it will be important to test this set of SSRs in more blueberry varieties.

These data (namely the specific positive markers) can support the search for strategies for the marker-assisted identification of genotypes. This identification is the key to the conservation of genetic resources and to helping farmers and breeders with different practical approaches to selection that could be useful in improving blueberry production. It will be important to expand and validate these specific markers in other genotypes from these species to prove their efficiency in cultivar characterization.

## Figures and Tables

**Figure 1 plants-13-03488-f001:**
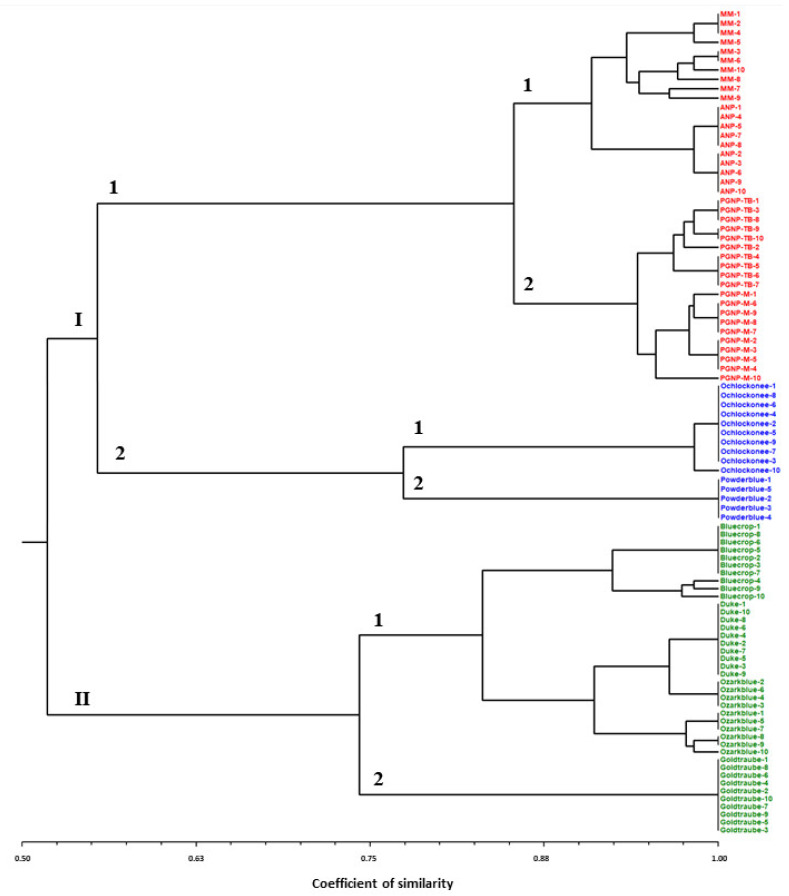
SSR-based dendrogram of the 95 genotypes belonging to three *Vaccinium* species constructed using unweighted pair-group arithmetic average (UPGMA) and similarity matrices (red—*V. myrtillus*; blue—*V. ashei*; and green—*V. corymbosum*).

**Figure 2 plants-13-03488-f002:**
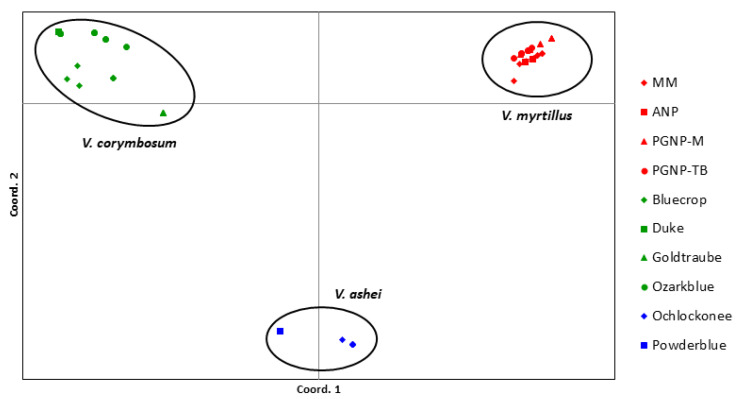
PCoA analysis of three *Vaccinium* species based on the genetic similarity matrix obtained with the eight SSR loci.

**Figure 3 plants-13-03488-f003:**
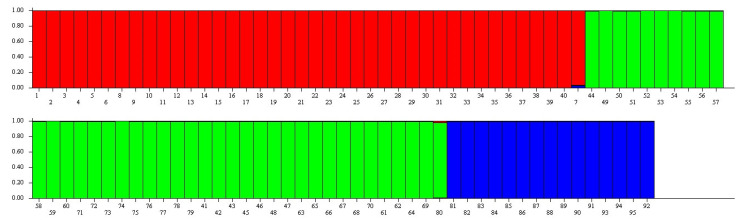
Assignment of three *Vaccinium* species by STRUCTURE for SSR markers. Each individual bar represents a sample/accession. Numbers, 1–40—*V. myrtillus* (red); 41–80—*V. corymbosum* (green) and; 81–95—*V. ashei* (blue).

**Table 1 plants-13-03488-t001:** Allelic diversity among the three *Vaccinium* species for eight simple sequence repeat (SSR) loci.

SSR Marker	Ranged Size (bp)	Total Alleles	Polymorphic Alleles	Unique Alleles	Rare Alleles	PIC
CA23F	151–168	8	8	1	1	0.29
CA112F	144–185	10	10	2	1	0.32
CA169F	113–165	8	8	1	0	0.29
CA190R	240–246	2	2	0	0	0.31
CA483F	295–376	14	14	4	2	0.37
CA94F	380–396	7	7	4	3	0.34
VCC_B3	225–253	4	4	1	0	0.28
VCC_I8	156–165	4	4	1	0	0.30
Total	-	57	57	14	7	0.32

PIC—polymorphism information content.

**Table 2 plants-13-03488-t002:** Genetic parameters analyzed in the three *Vaccinium* species and their cultivars/populations.

	Cult./Pop.	NG	NA	P. loci	EA	Ne
*V. myrtillus*	MA	10	12	7	0	1.079
	ANP	10	10	1	0	1.018
	PGNP-M	10	13	4	0	1.036
	PGNP-TB	10	13	3	0	1.028
	Average		12.0	3.75	0	1.040
*V. corymbosum*	Bluecrop	10	23	5	1	1.055
	Duke	10	22	0	0	1.000
	Goldtraube	10	18	0	2	1.000
	Ozarkblue	10	23	5	0	1.065
	Average		21.5	2.5	0.75	1.03
*V. ashei*	Ochlockonee	10	16	1	3	1.004
	Powderblue	5	19	0	6	1.000
	Average		17.5	0.5	4.5	1.002

MA—Marão mountain; ANP—Alvão Natural Park; PGNP-M—Peneda-Gêres National Park, Melgaço region; PGNP-TB—Peneda-Gêres National Park, Terras de Bouro region; Cult./Pop.—cultivar/population; NG—number of genotypes; NA—number of alleles per cultivar/population; P. loci—polymorphic loci; EA—exclusive alleles per cultivar/population; Ne—number of effective alleles.

**Table 3 plants-13-03488-t003:** Analysis of molecular variance (AMOVA) for the *V. ashei*, *V. corymbosum*, and *V. myrtillus* cultivars/populations.

	d.f.	MS	Variance Components	Total Variance	Value
Among populations	9	97.47	10.23	95%	0.949
Within populations	85	0.55	0.55	5%	*p* < 0.001

**Table 4 plants-13-03488-t004:** Summary information about the cultivars and populations used in this study.

Species	Cultivar or Population	Pedigree Information	Geographical Location of Collection
*V. corymbosum*	“Bluecrop”	GM-37 (Jersey × Pioneer) × CU-5 (Stanley × June)	Sever do Vouga, Aveiro, Portugal
	“Duke”	(Ivanhoe × Earliblue) × 192-8 (E-30 × E-11)	Sever do Vouga, Aveiro, Portugal
	“Goldtraube”	German variety	Sever do Vouga, Aveiro, Portugal
	“Ozarkblue”	G-144 × Florida 4-76	Sever do Vouga, Aveiro, Portugal
*V. ashei*	“Ochlockonee”	Tifblue (Ethel × Clara) × Menditoo (Myers × Black Giant)	Sever do Vouga, Aveiro, Portugal
	”Powderblue”	Tifblue (Ethel × Clara) × Menditoo (Myers × Black Giant)	Sever do Vouga, Aveiro, Portugal
*V. myrtillus*	ANP		Alvão, Vila Real, Portugal
	MM		Campeã, Vila Real, Portugal
	PGNP-M		Melgaço, Viana do Castelo Portugal
	PGNP-TB		Terras de Bouro, Braga, Portugal

**Table 5 plants-13-03488-t005:** The characteristics of the eight SSR loci analyzed. Repeat motifs, primer sequences (F = forward; R = reverse), annealing temperature, size, and labeling. The sizes presented are those described in the original study [26].

SSR Marker	Repeat Motif	Sequence	Ta (°C)	Size	Labeling
CA23F	(AGA)_6_	F: GAG AGG GTT TCG AGG AGG AG R: GTT TAG AAA CGG GAC TGT GAG ACG	62	150–170	VIC
CA112F	(AG)_7_	F: TCC ACC CAC TTC ACA GTT CA R: GTT TAT TGG GAG GGA ATT GGA AAC	62	140–200	NED
CA169F	(GAT)_4_	F: TAG TGG AGG GTT TTG CTT GG R: GTT TAT CGA AGC GAA GGT CAA AGA	62	109–130	PET
CA190R	(TGC)_5_	F: TTA TGC TTG CCA TGG TGG TA R: TTG CGA AGG GAC CTA GTA GC	62	250–280	FAM
CA483F	(TC)_8_	F: GTC TTC CTC AGG TTC GGT TG R: GAA CGG CTC CGA AGA CAG	62	300–370	FAM
CA94F	(AG)_7_	F: CAC CCA TTT CAC GGA ATC TC R: GTT TAC TTG GTC GGG TGT TGT CTC	62	390–420	FAM
VCC_B3	(AG)_9_	F: CCT TCG ATC TTG TTC CTT GC R: GTT TGA TGC AAT TGA GGT GGA GA	62	250–275	PET
VCC_I8	(TG)_8_	F: TTC AGC ATT CAA TCC ATC CA R: GTT TCT CTT CTC CAA TCT CTT TTC CA	62	120–140	FAM

Locus name prefix indicates source: CA, EST libraries constructed from cold climates; VCC, genomic-enriched.

## Data Availability

Data are contained within the article.

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
