# Peer review of "Genetic Diversity and Identification of *Vaccinium* Species Through Microsatellite Analysis"

_plants, 2024, doi:10.3390/plants13243488_

Round 1
Reviewer 1 Report
Comments and Suggestions for Authors
The authors the genetic analysis was conducted on three Vaccinium species, for a total of 95 genotypes, using eight microsatellite (SSRs) loci were detected. Also the authors identified a set of SSR markers that can be used for cultivar and species identification in blueberries. Blueberry species are of commercial importance in world blueberry production. Blueberries are important in a healthy diet. Traditional methods of identification and characterization of blueberry varieties by morphological features are currently insufficient. This work is relevant. There are several comments about the work
Table 1.
- Exclusive alleles were detected in one genotype? Unique alleles - would be better. It would be good to detect the number of rare alleles occurring in no more than 5% of samples.
- It is necessary to rename the table column: primer to SSR marker. Remove P and F from the marker names, otherwise SSR markers may be taken for primers, which will be incorrect.
- How polymorphism was calculated?
- Calculate PIC for each SSR marker, it will be useful.
Figure 1.
- In Figure 1, the names of the genotypes are difficult to distinguish.
Discussion
The set of eight SSR primer pairs from this study was successfully amplified in three blueberry species. These species could be distinguished using 8 microsatellites. A total of 95 samples were analyzed.
- Were all samples distinguished using 8 markers?
It is not necessary to have unique alleles for each genotype; it is enough to have an individual spectrum of microsatellite amplicons in the samples.
Is it possible to distinguish more blueberry samples/species using this set? In this case, this work will be an approach to certify blueberry samples using microsatellites.
Author Response
Reviewer #1
The authors the genetic analysis was conducted on three Vaccinium species, for a total of 95 genotypes, using eight microsatellite (SSRs) loci were detected. Also the authors identified a set of SSR markers that can be used for cultivar and species identification in blueberries. Blueberry species are of commercial importance in world blueberry production. Blueberries are important in a healthy diet. Traditional methods of identification and characterization of blueberry varieties by morphological features are currently insufficient. This work is relevant. There are several comments about the work.
Authors: Thank you for the comments. All of them have been considered to improve the manuscript quality.
Table 1.
- Exclusive alleles were detected in one genotype? Unique alleles - would be better. It would be good to detect the number of rare alleles occurring in no more than 5% of samples.
Authors: We changed the exclusive alleles to unique alleles, and these alleles were considered when only was detected in one of the cultivars/populations. Changes were inserted in the main text, please see table 1 and lines 116.
- It is necessary to rename the table column: primer to SSR marker. Remove P and F from the marker names, otherwise SSR markers may be taken for primers, which will be incorrect.
Authors: We renamed the SSR marker, please see table 2, line 127. We put the exact name from the Boches et al. 2005 study, and they attributed that lettering from each locus. We decided to maintain the exact nomenclature of the reference to make easier the comparison with other studies.
- How polymorphism was calculated?
Authors: The % polymorphism is an error. It is not an important measurement.
- Calculate PIC for each SSR marker, it will be useful.
Authors: PIC values have been calculated and inserted in table 1. Also the information related to PIC has completed in section of material and methods. Please see lines, 116 and inserted in the material and methodology section.
Figure 1.
- In Figure 1, the names of the genotypes are difficult to distinguish.
Authors: Several improvements have been performed in the figure, we hope the quality is good now. Please see figure 1, line 161.
Discussion
The set of eight SSR primer pairs from this study was successfully amplified in three blueberry species. These species could be distinguished using 8 microsatellites. A total of 95 samples were analyzed.
- Were all samples distinguished using 8 markers? It is not necessary to have unique alleles for each genotype; it is enough to have an individual spectrum of microsatellite amplicons in the samples.
Authors: The 95 samples (40 from V. myrtillus, 40 from V. corymbosum and 15 from V. ashei) were used to this study. These 95 samples and 8 SSR primers were analyzed together being used to distinguish the three Vaccinium species. The unique alleles were considered the alleles that were only present in one cultivar or population, being in the future potential markers for the identification of the cultivars/populations. We inserted new information’s in the manuscript to clarify this point.
Is it possible to distinguish more blueberry samples/species using this set? In this case, this work will be an approach to certify blueberry samples using microsatellites.
Authors: Blueberry producers has a high difficulty in varieties identification using only the morphological characteristics, for this reason we consider important to screening new approaches for blueberry identification. The objective is to implement this set of primers in the future for the identification of blueberry varieties. This opportunity has been included in final conclusion, please see lines 419-423.
Reviewer 2 Report
Comments and Suggestions for Authors
The study's conceptual design is well structured and interesting, highlighting aspects of a genetically undervalued species. However, the manuscript's writing is sometimes imprecise and contradictory.
Here are my comments:
Introduction: The section on commercial/economic importance of Vaccinium cultivars could be expanded (with some examples). Furthermore, it would be appropriate to mention the domestication process of the wild species V. myrtillus.
Materials and methods: for plant material, a genotype-species-geographical location summary table would be helpful.
Line 265-269: Revise and correct the sentence ‘...the Terras de Bouro region (PGNP-TB). Each cultivar/population...’.
Results
Table 1: Is it possible to calculate and enter the PIC value (Polymorphism Information Content) for each assayed locus?
In order to make the paper more readable, it would be appropriate to give the mean values for each species and the overall mean of the three species in Table 2.
Figure 1. The phylogenetic tree shows only two clusters. In cluster 1 V. myrtillus and V. ashei genotypes are grouped and separated into two subclusters.
Discussions
Line 167: Explain the concept of 'exclusive alleles'. In addition, the sentence 'A number of exclusive alleles have been detected' should be moved to line 175.
Line 179: In my opinion, the lack of detection of exclusive alleles is due to the SSR markers used (low specificity and polymorphism for the V. myrtillus species) and not to environmental conditions.
Line 183: The text first states that the set of SSR markers used does not allow to detect exclusive alleles in V. myrtillus, but later states that three were observed.
Comments on the Quality of English LanguageSentences too long and articulate, that make the paper less fluent. It is recommended a manuscript revision for more fluent English.
Author Response
Reviewer #2
The study's conceptual design is well structured and interesting, highlighting aspects of a genetically undervalued species. However, the manuscript's writing is sometimes imprecise and contradictory.
Authors: Thank you for the comments. Several improvements have been performed in all the manuscript.
Introduction: The section on commercial/economic importance of Vaccinium cultivars could be expanded (with some examples). Furthermore, it would be appropriate to mention the domestication process of the wild species V. myrtillus.
Authors: Text about Vaccinium varieties have been inserted in the introduction section, please see lines 32-33, 37-47, 50-67.
Materials and methods: for plant material, a genotype-species-geographical location summary table would be helpful.
Authors: A summary table was inserted in the main text, and the figure was eliminated to make clearer this section. Please see, table 4 and line 328.
Line 265-269: Revise and correct the sentence ‘...the Terras de Bouro region (PGNP-TB). Each cultivar/population...’.
Authors: Thank you for noticing the typo. The text has been changed.
Results
Table 1: Is it possible to calculate and enter the PIC value (Polymorphism Information Content) for each assayed locus?
Authors: PIC values have been calculated and inserted in table 1. Also the information related to PIC has completed in section of material and methods. Please see lines, 116 and inserted in the material and methodology section.
In order to make the paper more readable, it would be appropriate to give the mean values for each species and the overall mean of the three species in Table 2.
Authors: Changes in table 2 has been performed, please see table 2, line 127.
Figure 1. The phylogenetic tree shows only two clusters. In cluster 1 V. myrtillus and V. ashei genotypes are grouped and separated into two subclusters.
Authors: The phylogenetic tree was changed and the main text was also changed to achieve all the comments performed by the four reviewers, please see figure 1 line 161.
Discussions
Line 167: Explain the concept of 'exclusive alleles'. In addition, the sentence 'A number of exclusive alleles have been detected' should be moved to line 175.
Authors: We changed the ‘exclusive’ to ‘unique’. Information in the main text were inserted on the main text to clarify the results. Please see lines
Line 179: In my opinion, the lack of detection of exclusive alleles is due to the SSR markers used (low specificity and polymorphism for the V. myrtillus species) and not to environmental conditions.
Authors: Thank you for the comment. Several informations have been inserted in the main text. Somes reports have referred the Portuguese populations of V. myrtillus are isolated low genetic diverse at the limit of the range of distribution, face to its current biogeographic regression (Tikhomirov et al. 2022), as well as the lack of taxonomic diversity to promote genetic flow (Bjedov et al. 2015). This is an usual phenomenon observed for other different taxa, where not sufficiently differences between populations were not wide enough to separate and identify exclusive alleles (Alsos et al. 2012; Zhang et al. 2020). Please see lines 221-233.
Line 183: The text first states that the set of SSR markers used does not allow to detect exclusive alleles in V. myrtillus, but later states that three were observed.
Authors: We identified three unique alleles specific of V. myrtillus species, but when we did not identified unique alleles in V. myrtillus populations. Text has been changed to make this idea clearer. Please see lines 221-233.
Reviewer 3 Report
Comments and Suggestions for Authors
After minor corrections, the manuscript can be published.

Author Response
Reviewer #3
Line 124 - Correct the name of the variety 'Powderblue'
Authors: Thank you. The figure legend has been corrected. Please see figure 1, line 161.
Lines 206 – 208 - Do the authors believe that only the indicated environmental factors are responsible for the differences?
Authors: Thank you for the comment. Several informations have been inserted in the main text. Some reports have referred the Portuguese populations of V. myrtillus are isolated low genetic diverse at the limit of the range of distribution, face to its current biogeographic regression (Tikhomirov et al. 2022), as well as the lack of taxonomic diversity to promote genetic flow (Bjedov et al. 2015). This is an usual phenomenon observed for other different taxa, where not sufficiently differences between populations were not wide enough to separate and identify exclusive alleles (Alsos et al. 2012; Zhang et al. 2020). Please see lines 221-233.
Lines 216 – 220 - In the introduction, the authors emphasize "Traditional methods of cultivar identification and characterization in blueberries are based on morphological traits, which are heavily influenced by environmental conditions and production practices. Additionally, collecting these traits can be time-consuming, especially when surveying large populations in different locations" and therefore it is better to use DNA markers. In this fragment, however, they state that morphological features are responsible for the distinct grouping of genotypes. The reasons for the differentiation should be sought at the molecular level, paying attention to, for example, the pedigrees of cultivars.
Authors: This sentence is related to wild blueberries and we did not have information about the pedigrees. The only difference between them is their locations and environment.
Lines 225-226 - What causes caused the joint grouping of some accessions 'Ozarkblue' and 'Duke' and cv's.
Authors: Thank you for the comment. We collected the samples in local farmers, that have several years of experience and buy the plants to certified producers, however some mistakes can be occur during this process.
Line 294 - Explain the abbreviation.
Authors: The information has been inserted in the main text. Please see lines 357.
Reviewer 4 Report
Comments and Suggestions for Authors
Provided manuscript describes application of eight SSR markers on 95 samples of three Vaccinium species in order to test cross-species transferability and genetic diversity. Current version of the manuscript contains some flaws, mainly in the Results and Discussion sections, but I believe the most of them can be corrected.
The other thing is the real importance for blueberries breeding. I can not get rid of the impression that in the era of genomics and phenomics is the impact of presented study limited due to low number of used markers and genotypes for this kind of research.
1) Explanation of experimental design
I would like to have better understanding of presented experimental design. The authors mentioned several times analysis of 95 genotypes (line 17, 77, 86, 162), but should not be the plants belonging to the given cultivar genetically the same? Please, correct me if I am wrong, but for breeding of highly allogamous species is typical crossing between suitable parental genotypes and selected hybrids are propagated only vegetatively. Therefore all plants are technically clones i.e. genetically identical. I was not able to find detailed information
about breeding of blueberries, but my assumption was partially confirmed in the summary table 2 (line 91), but there was surprisingly high level of polymorphism within ‘Bluecrop’ and ‘Ozarkblue’ cultivars. Why the authors tested 10 plants per cultivar if my assumption about cultivar breeding is correct? Would not be better to test more cultivars and less plants per cultivar? Anyway, the number of genotypes is probably substantially lower than presented.
Another thing related to experimental design are markers of choice. Why EST-SSR/gSSR markers? One of their greatest advantage, codominance, was eliminated in behalf of proper statistical analysis and this decision declined information value per locus. Therefore, this should be compensated by more markers. Why was tested only 10 SSR markers? I made some quick searching and the number of available SSR markers is several times higher. Or would not be better use RAD-Seq or AFLP method in this case? Nonetheless, the
key problematic aspect is the very low number of markers.
I think it was not lucky idea made cross-species transferability test with combination of diversity study and partially population genetics study. Different kind of genetic signals are mixed together and made data
interpretation more difficult. Of course, low number of markers made weaknesses of this approach more obvious.
2) More information about biology of studied species
In comparative study of different species I would expect more information about biology of each species like mode of reproduction, level of ploidy, some morphological characteristics or even picture of each species. It is difficult to morphologically recognize V. corymbosum, V. Ashei R. and V. myrtillus L. from each other? Some biological information are provided only for V. myrtillus (line 35 - 36).
3) Materials and Methods - Plant Material
Providing map of localities (line 274) is good idea, but there is no scale. It is hard to estimate distances without proper scale.
4) Materials and Methods - DNA Extraction
All (minor or not) modifications of DNA extraction protocol should be described (line 281).
5) Materials and Methods - PCR
It was really used 1 µL of 10µM primers (i.e. 1 µM) in the reaction mixture (line 297)? It seems to be unusually high (we typically use 0.1 - 0.2 µM per reaction in our labs).
Presented analysis comprises of following steps: a) multiplex PCR with fluorescently labeled primers, b) test of amplification result via agarose gel electrophoresis and c) analysis via capillary electrophoresis. I do not understand why was made intermediate step with agarose gel electrophoresis? Was it part of optimisation procedure, and then it should be mentioned, or was is really done for whole collection of samples?
6) Materials and Methods - Data Analysis
What does it mean “polymorphic allele” (line 315-316)? It means fragments of the same size but different sequence or what?
Does it make a sense calculate effective number of alleles for binary scored markers (line 317-319)? Why were calculated Shannon Information Index, diversity (h) and unbiased diversity (uh)? There is neither information nor commentary except sentences like “highest value of this parameter and lowest value of that parameter” and that is all. It is like numbers for numbers without any biological meaning.
7) Results - The number of alleles
How it is possible to have “. . . a total of 57 different alleles across the 95 genotypes (Table 1)” (line 77-78) with mean 7.125 alleles per locus and “. . . a total of 169 alleles for the 95 genotypes (Table 2) and an average of 21.125 alleles per locus” (line 86) at once? There must be something wrong. I suppose the answer provides incorrectly interpreted Table 2 (line 91). The column “NA” shows number of alleles for given cultivar/population, but it is not “the number of alleles/locus” (line 84). I presume this is the number of detected alleles for all loci together for given cultivar/population. Moreover, the values in the “NA” column can not be simply summed up because some alleles are shared across cultivars or populations. Therefore, shared alleles would be counted two or multiple times. Probably, this is the source of the discrepancies between 57 alleles (line 77) and 169 alleles (line 86).
8) Results - Polymorphism
What means last column “% Polymorphism” in the Table 1 (line 82)? Locus is polymorphic in case of two or more alleles were detected (some researches combine it with another criterion using minimal allelic frequency 1% or 5%). Average polymorphism is defined as a number of polymorphic loci over all tested loci, but 100% polymorphism for each locus? What would be meaning of 50% polymorphism? Please explain it.
9) Results - AMOVA
My opinion is quite opposite to the authors interpretation of AMOVA result (line 113 - 115). This interpretation supports Table 3 (line 116) where is 95% of total variance among populations (i.e. cultivars/populations/species) and only 5% of total variance is withing populations. It is also supported by the results of used multivariate techniques (see cluster analysis, PCoA, STRUCTURE - line 121, 141, 153).
P-value for ΦP T parameter is different in the main text (line 113) and Table 3 (line 116) i.e. P<0.0001 vs P<0.001.
10) Results - STRUCTURE
What K value was really set in the STRUCTURE? In the Results section is written “SSR data were analysed by increasing the number of subpopulations (K) from 1 to 7.” (line 145 - 146), but different value is mentioned in the Materials and Methods section where is written “. . . using 1,000,000 MCMC repetitions and 50,000 burn-in periods be setting K from 1 to 5.” (line 335 - 336. Anyway, it looks like arbitrary values. Why was K set to 5 or 7 and not 10 (or 11)?
I believe graphs with estimation of K values via lnPr and ∆K methods would be great for the illustration of the results.
11) Discussion
Related parts of the discussion must be changed in order to reflect the changes in the Results and Material and Methods sections.
What is the authors’ explanation of higher genetic variability within the samples of the same cultivar (‘Ozarkblue’, ‘Bluecrop’) than in the wild population (ANP) of V. myrtillus? Was the identity of cultivars verified somehow?
Substantial portion of Discussion section tries to explain observed genetic differences between the populations of V. myrtillus by climatic or environmental conditions (line 187 - 212). Similarly, observed clustering of
V. ashei cultivars is explained by different morphological characteristics (line 213 - 220). Such statements are very vague and general and what is more important, they do not have any support in the presented data. The authors did not mention any functional or association study nor provide any information about involved genes. Moreover, it is a little bit ironic when sentences from Discussion section are compared with sentences about molecular markers in the Introduction section (for example lines 46-47 “. . . since they [molecular markers] show high level of polymorphisms and are not affected by environmental factors [9].”).
12) Minor issues
“In the eight SSR loci analysed, 57 different alleles across the 95 genotypes were detected (Table 2).” (line 162 - 163) -> It is Table 1 not Table 2.
“Beyond the efficacy of cross-species transferability of SSR sequences in this study, they also allowed the differentiation of sweet cherry genotypes.” (line 244 - 245) -> ???
“each cultivar/population 10 genotypes. . . ” (line 267) -> Each cultivar/population . . .
Author Response
Reviewer #4
Provided manuscript describes application of eight SSR markers on 95 samples of three Vaccinium species in order to test cross-species transferability and genetic diversity. Current version of the manuscript contains some flaws, mainly in the Results and Discussion sections, but I believe the most of them can be corrected.
The other thing is the real importance for blueberries breeding. I can not get rid of the impression that in the era of genomics and phenomics is the impact of presented study limited due to low number of used markers and genotypes for this kind of research.
Authors: Thank you for your comment. We acknowledge your observation regarding the Results and Discussion sections, and several changes have been performed. Regarding the limited number of molecular markers, we understand your concerns, but our study as designed as an initial step to evaluate cross-species transferability of SSR markers and their utility in assessing genetic diversity within and across Vaccinium species. We considered important due the importance of wild blueberry that currently is in biogeographic regression due the thermal recovery that the planet is developing. This situation makes the species in question gain even more relevance, as this movement northwards makes the morphogenomes more resistant towards the end. Precisely the results of the article show these differences, in this case genetic, that better characterize this biogeographic phenomenon. We agree that the advances in genomics and phenomics offer a powerful tool, but SSR markers remain relevant and the cost benefits were important. Our results demonstrate that the SSR markers can still provide valuable insights into genetic relationships and diversity in Vaccinium species, supporting their continued utility in specific breeding or conservation efforts. Despite the limited number of markers, our study provides important data on cross-species marker transferability, which is particularly relevant for researchers working on related but under-studied Vaccinium species. Moreover, the observed genetic diversity patterns contribute to a better understanding of the genetic resources available for breeding programs. These findings, we believe, lay the groundwork for future studies incorporating high-throughput genomic techniques and larger sample sizes. We have revised the manuscript to better articulate these points in both the introduction and discussion sections.
1) Explanation of experimental design
I would like to have better understanding of presented experimental design. The authors mentioned several times analysis of 95 genotypes (line 17, 77, 86, 162), but should not be the plants belonging to the given cultivar genetically the same? Please, correct me if I am wrong, but for breeding of highly allogamous species is typical crossing between suitable parental genotypes and selected hybrids are propagated only vegetatively. Therefore all plants are technically clones i.e. genetically identical. I was not able to find detailed information
about breeding of blueberries, but my assumption was partially confirmed in the summary table 2 (line 91), but there was surprisingly high level of polymorphism within ‘Bluecrop’ and ‘Ozarkblue’ cultivars. Why the authors tested 10 plants per cultivar if my assumption about cultivar breeding is correct? Would not be better to test more cultivars and less plants per cultivar? Anyway, the number of genotypes is probably substantially lower than presented.
Authors: Thank you for comment. Yes, blueberries were propagated vegetatively, as other allogamous species, and is expected that populations will be genetically identical. The identification of these cultivars is performed mainly by morphological traits, but some factors can contribute to genetic variability, as:
- somatic mutation may occur during the propagation process
- environmental conditions or methodology for micropropagation
- and an incorrect morphological characterization during the process.
In our study we observed higher polymorphism than the expected on ‘Bluecrop’ and ‘Ozarkblue’ cultivars, and can be related with clonal variation or wrong identification of cultivars. Following the criterion of other studies (e.g. Bassil et al. 2020) we decided Testing 10 plants per cultivar. We considered a good choice to account for possible intra-cultivar variability, ensuring robust statistical power and minimizing the impact of outliers. This decision also allowed us to better understand the potential genetic diversity within widely cultivated blueberries. The number of genotypes refers to the distinct plants analyzed, including individuals from different cultivars. While it is true that plants within a single cultivar are often considered clones, this term encompasses the total plants sampled across all cultivars. The observed intra-cultivar variation suggests that genetic uniformity in some cultivars may not be absolute, warranting the inclusion of multiple plants per cultivar. Relatively to testing more cultivars than plants per cultivar, the scope of our study was focused on understand the diversity from different cultivars high higher expression in Portugal and exploit the diversity of wild blueberries. We have revised the manuscript to clarify these aspects in the methods and discussion sections.
Another thing related to experimental design are markers of choice. Why EST-SSR/gSSR markers? One of their greatest advantage, codominance, was eliminated in behalf of proper statistical analysis and this decision declined information value per locus. Therefore, this should be compensated by more markers. Why was tested only 10 SSR markers? I made some quick searching and the number of available SSR markers is several times higher. Or would not be better use RAD-Seq or AFLP method in this case? Nonetheless, the
key problematic aspect is the very low number of markers.
I think it was not lucky idea made cross-species transferability test with combination of diversity study and partially population genetics study. Different kind of genetic signals are mixed together and made data
interpretation more difficult. Of course, low number of markers made weaknesses of this approach more obvious.
Authors: Thank you for comments. We chose gSSR markers primarily because they are cost-effective, relatively simple to implement, and provide sufficient resolution for the objectives of our study. We understand that codominance was not utilized to its fullest due to the statistical analysis method we employed, the chosen approach allowed us to focus on other important aspects, such as marker polymorphism and reliability. Regarding the number of markers, we understand that using only 8 SSR markers may limit the resolution and information obtained. However, 8 SSR markers are in agreement and proven effectiveness in studies involving blueberry and related species (Debnath et al. 2014; Bassil et al. 2020; Bidani et al. 2017). We acknowledge that increasing the number of markers or using advanced approaches, such as RAD-Seq or AFLP, could improve the analysis, but these methods require greater financial and computational resources, which were beyond the scope of this study. In future studies, we will consider the integration of a higher number of markers or explore next-generation sequencing approaches.
2) More information about biology of studied species
In comparative study of different species I would expect more information about biology of each species like mode of reproduction, level of ploidy, some morphological characteristics or even picture of each species. It is difficult to morphologically recognize V. corymbosum, V. Ashei R. and V. myrtillus L. from each other? Some biological information are provided only for V. myrtillus (line 35 - 36).
Authors: The introduction section was completed with information about the three species under study, please see lines xxx.
3) Materials and Methods - Plant Material
Providing map of localities (line 274) is good idea, but there is no scale. It is hard to estimate distances without proper scale.
Authors: Thank you for comments. For suggestion of other revision, we changed the figure to a summary table with the aim to make clearer the information. Please see line xxx.
4) Materials and Methods - DNA Extraction
All (minor or not) modifications of DNA extraction protocol should be described (line 281).
Authors: Thank you for comment. We inserted in the main text the reference for the exact minor modifications. Please see line xxx.
5) Materials and Methods - PCR
It was really used 1 µL of 10µM primers (i.e. 1 µM) in the reaction mixture (line 297)? It seems to be unusually high (we typically use 0.1 - 0.2 µM per reaction in our labs).
Presented analysis comprises of following steps: a) multiplex PCR with fluorescently labeled primers, b) test of amplification result via agarose gel electrophoresis and c) analysis via capillary electrophoresis. I do not understand why was made intermediate step with agarose gel electrophoresis? Was it part of optimisation procedure, and then it should be mentioned, or was is really done for whole collection of samples?
Authors: Thank you for comments. To make clear the methodology, we reformulate this information in the manuscript. The agarose gel electrophoresis was performed as validation of the PCR amplification for the following analysis via capillary electrophoresis. Please see lines xxx.
6) Materials and Methods - Data Analysis
What does it mean “polymorphic allele” (line 315-316)? It means fragments of the same size but different sequence or what?
Does it make a sense calculate effective number of alleles for binary scored markers (line 317-319)? Why were calculated Shannon Information Index, diversity (h) and unbiased diversity (uh)? There is neither information nor commentary except sentences like “highest value of this parameter and lowest value of that parameter” and that is all. It is like numbers for numbers without any biological meaning.
Authors: Thank you for comments. We accepted your suggestion, and these parameters were eliminated from the table. Please see, lines xxx.
7) Results - The number of alleles
How it is possible to have “. . . a total of 57 different alleles across the 95 genotypes (Table 1)” (line 77-78) with mean 7.125 alleles per locus and “. . . a total of 169 alleles for the 95 genotypes (Table 2) and an average of 21.125 alleles per locus” (line 86) at once? There must be something wrong. I suppose the answer provides incorrectly interpreted Table 2 (line 91). The column “NA” shows number of alleles for given cultivar/population, but it is not “the number of alleles/locus” (line 84). I presume this is the number of detected alleles for all loci together for given cultivar/population. Moreover, the values in the “NA” column can not be simply summed up because some alleles are shared across cultivars or populations. Therefore, shared alleles would be counted two or multiple times. Probably, this is the source of the discrepancies between 57 alleles (line 77) and 169 alleles (line 86).
Authors: Thank you for comments. Changes were performed in table and text. Please see lines xxx.
8) Results - Polymorphism
What means last column “% Polymorphism” in the Table 1 (line 82)? Locus is polymorphic in case of two or more alleles were detected (some researches combine it with another criterion using minimal allelic frequency 1% or 5%). Average polymorphism is defined as a number of polymorphic loci over all tested loci, but 100% polymorphism for each locus? What would be meaning of 50% polymorphism? Please explain it.
Authors: Thank you for comments. To answer the suggestions of another reviewer. Table 1 has been modified accordingly. Please see table 1, line 116.
9) Results - AMOVA
My opinion is quite opposite to the authors interpretation of AMOVA result (line 113 - 115). This interpretation supports Table 3 (line 116) where is 95% of total variance among populations (i.e. cultivars/populations/species) and only 5% of total variance is withing populations. It is also supported by the results of used multivariate techniques (see cluster analysis, PCoA, STRUCTURE - line 121, 141, 153).
P-value for ΦP T parameter is different in the main text (line 113) and Table 3 (line 116) i.e. P<0.0001 vs P<0.001.
Authors: Thank you for comments. Sorry has been a mistake. We agree you’re your interpretation. Please see lines 150-152.
11) Discussion
Related parts of the discussion must be changed in order to reflect the changes in the Results and Material and Methods sections.
What is the authors’ explanation of higher genetic variability within the samples of the same cultivar (‘Ozarkblue’, ‘Bluecrop’) than in the wild population (ANP) of V. myrtillus? Was the identity of cultivars verified somehow?
Substantial portion of Discussion section tries to explain observed genetic differences between the populations of V. myrtillus by climatic or environmental conditions (line 187 - 212). Similarly, observed clustering of
V. ashei cultivars is explained by different morphological characteristics (line 213 - 220). Such statements are very vague and general and what is more important, they do not have any support in the presented data. The authors did not mention any functional or association study nor provide any information about involved genes. Moreover, it is a little bit ironic when sentences from Discussion section are compared with sentences about molecular markers in the Introduction section (for example lines 46-47 “. . . since they [molecular markers] show high level of polymorphisms and are not affected by environmental factors [9].”).
Authors: Thank you for comments. This study is a first approach that we performed using SSR markers, in the future we intend to explore more this thematic and if possible study some genes that may be involved in these differences. The cultivars of V. corymbosum and V. ashei were obtained in local producers and during the planting process, errors in identification may have occurred, or a clonal variation inherent to the vegetative propagation process. However, some reports point some hypothesis about the effect of ploidy or the method type in obtaining hybrid cultivars that can generate spontaneous in vivo mutation and misidentified cultivars (Bassil et al. 2020; Miteca et al. 2024). For other hand, in V. myrtillus, the genotypes were not result from vegetative propagation, they are natural populations. Some reports have referred the Portuguese populations of V. myrtillus are isolated low genetic diverse at the limit of the range of distribution, face to its current biogeographic regression (Tikhomirov et al. 2022), as well as the lack of taxonomic diversity to promote genetic flow (Bjedov et al. 2015). This is an usual phenomenon observed for other different taxa, where not sufficiently differences between populations were not wide enough to separate and identify exclusive alleles (Alsos et al. 2012; Zhang et al. 2020).
12) Minor issues
“In the eight SSR loci analysed, 57 different alleles across the 95 genotypes were detected (Table 2).” (line 162 - 163) -> It is Table 1 not Table 2.
“Beyond the efficacy of cross-species transferability of SSR sequences in this study, they also allowed the differentiation of sweet cherry genotypes.” (line 244 - 245) -> ???
“each cultivar/population 10 genotypes. . . ” (line 267) -> Each cultivar/population . . .
Authors: Thank you for comments. These minor issues have been corrected.
Round 2
Reviewer 4 Report
Comments and Suggestions for Authors
Commentary and suggestions for the authors are part of attached file.

Author Response
I have received a revised version of the manuscript despite my rejection decision, but I respect editors' opinion and (maybe) other reviewers. Due to time pressure I am unable to do a proper review of the revised manuscript. Therefore, I have only checked highlighted changes. I appreciate the authors improved overall quality of the manuscript though I am still not convinced whether analysis with such a low number of markers should be published. I do understand arguments of the authors, but I cannot judge the current manuscript through the lens of a future (planned?) project.
Authors: Thank you for all the comments performed to the manuscript. They were very important for the manuscript improvement.
- Introduction section
Introduction part was improved and more information about the studied species was added.
Authors: Thank you for the comment.
- Results section
Now I understand what was meant by sentence “… a total of 169 [revised to 149] alleles for the 95 genotypes (Table 2)”, but I think it could be confusing for the reader.
It seems the resolution of Figure 1 [line 161] is lower than previously and now it is difficult to read any description. I hope the resolution would be higher in the final version.
Authors: Thank you for the comments. We improve the sentence and the figure, please see lines 114-115 and 161.
I still do not know why K was tested from 1 to 7 [line 176]?
Authors: Thank you for the comment. In a first time, we decided to test 1 to 7 in comparison to other studies, this was a aleatory selection. To answer to the reviewer, we performed a new analysis testing the K 1 to 10, and the result were similar. However, one of the software is not available in the moment. So, we considered to maintain the structure analysis preformed initially.
- Discussion section
Explanation of “polymorphic allele” was provided.
The total number of alleles was changed in the Results section “… with a total of 149 alleles for the 95 genotypes (Table 2) and an average of 16.9 alleles per locus.” [line 116 - 117], but remained the same in the Discussion section “In this study a total of 169 alleles for the 95 genotypes and an average of 21.125 alleles per locus were observed.” [line 202 - 203].
Authors: Thank you for the comment. We corrected the sentence in the discussion section, it was a mistake. Please see line 203-204.
The authors suggested an explanation for unusual high intra-cultivar variability.
I still do not agree with arguments about genetic differentiation caused by climatic conditions because there is not solid support in the presented data and the results. It is more opinion than scientific conclusion.
Authors: Thank you for the comment. We changed the sentence and reformulate the text. Please see lines 218-235. Our objective was to propose a hypothesis for the intra-population/cultivar variability and not a scientific conclusion. However, several studies pointed edaphon-climatic conditions as an influence on genetic diversity in other crops.
- Material and Methods section
Unclear parts of the described procedure were changed and small errors corrected.
Redundant descriptive parameters were removed.
Authors: Thank you for the comment.
